# Unlocking Digitalization in Forest Operations with Viewshed Analysis to Improve GNSS Positioning Accuracy

Eugene Lopatin [1,*], Kari Väätäinen [1], Antero Kukko [2], Harri Kaartinen [2], Juha Hyyppä [2], Eero Holmström [1], Lauri Sikanen [1], Yrjö Nuutinen [1] and Johanna Routa [1]

1. Natural Resources Institute Finland, Luke, 80101 Joensuu, Finland
2. Finnish Geospatial Research Institute FGI, National Land Survey of Finland, Vuorimiehentie 5, 02150 Espoo, Finland
* Correspondence: eugene.lopatin@luke.fi; Tel.: +358-50-390-5002

**Abstract:** This study evaluated the positioning accuracy of moving forest harvesters using global navigation satellite system (GNSS) signals under a forest canopy, and developed approaches for forecasting accuracy under a mature spruce canopy cover. Real-time kinematic positioning with a Trimble R12 receiver on top of a harvester achieved high positioning accuracy, with 86% of observations meeting a maximum precision of 8 mm. However, the presence of a canopy cover hampered the GNSS's performance as there were fewer satellites available, leading to an increased number of inaccurate positions and larger values of the dilution of precision in geometry (GDOP), position (PDOP), vertical (VDOP) and horizontal directions (HDOP). The canopy cover estimated from the viewshed analysis of the digital surface model (DSM) was found to be a significant predictor of the dilution of precision and maximum deviation from the true position. These findings suggest that viewshed analysis provides more precise results than a mere canopy cover percentage for evaluating the impact of canopy cover on the GNSS's positioning of a harvester, despite its computational demands. Developing intelligent algorithms for precise positioning under the canopy can facilitate autonomous harvesting and forwarding, allowing for the implementation of digitalization in forest operations.

**Keywords:** digitalization; forest operations; viewshed analysis; GNSS positioning accuracy; harvester; canopy cover; forecasting accuracy; autonomous harvesting

## 1. Introduction

Forestry plays a crucial role in the environment, the economy, and society [1]. However, decision-making in the sector has been hampered by a lack of adequate knowledge [2] and data, especially on the pre-planning of harvesting [3]. The existing forest data also fail to fully advance and utilize the benefits of digitalization and automation. Hence, there is a requirement to create innovative solutions to support the development of the industry towards precision forestry. One of the quickest ways to leverage digitalization in forest operations is through the development of autonomous harvesting and forwarding, which necessitates the creation of intelligent algorithms for precise positioning under the forest canopy.

Studies have shown the impact of canopy cover on the positional accuracy of global navigation satellite system (GNSS) devices [4–11]. However, these results are limited to test environments and do not account for the effects of harvester operations or the capabilities of new GNSS devices. A recent study [11] emphasized the main advantages of precision forestry to reduce operational costs and yield a reduced negative environmental impact. However, any inaccuracies in the positioning of forestry machinery will lead to increased costs and an increased carbon footprint.

Harvesters are now collecting large amounts of log data during tree harvesting at the individual tree level based on sensors on the harvester head [12]. These data have an economic value and can provide unprecedented information on the characteristics of

individual trees and the relationships among trees, forests, and their environment. This information can help those involved in the forestry sector to make better models and decisions that support the environment, the economy, and society. Many harvesters are equipped with GNSS devices, allowing the data to be georeferenced. However, the accuracy of this georeferencing is often uncertain, and implementations are often sub-optimal; it can be influenced by factors such as canopy cover which breaks the line of sight to navigation satellites. There is a need to establish a method for evaluating the harvester GNSS's positioning, especially under a mature canopy cover, so that the information of the canopy cover can be used to predict the expectation for the GNSS's positional accuracy.

Conventional even-aged rotational forest management concludes in clear cutting and forest regeneration by natural, sowing, or planting [13–15]. Continuous cover forestry is a silvicultural alternative aimed at achieving multifunctional objectives including increased species diversity, a more resilient stand structure, and increased carbon storage [16–21]. Improving the accuracy of under-canopy harvester positioning is a necessity to increase the efficiency and practicality of this approach, including the development of new harvesting methods based on the precise positioning, tree selection, and collection of stand structure data before and after harvesting.

An accurate harvester positioning under the canopy can lead to several benefits: an improved efficiency, better harvest quality and improved forest growth potential, enhanced safety, reduced negative environmental impact and waste, and better inventory management. Recent studies which utilized low-cost mobile devices [11] were conducted in stands without GNSS devices on the harvesters [4–7,9–11]. Upgrading GNSS receivers with advanced technologies such as real-time kinematic (RTK) or differential GNSSs can improve accuracy. Using multiple GNSS constellations, such as global positioning system (GPS), globalnaja navigatsionnaja sputnikovaja sistema (GLONASS), and Galileo, can also increase accuracy by providing additional signals from different directions.

The objective of this study was to evaluate the positional accuracy of operating a harvester based on a modern GNSS device logging multiple GNSS constellations under a mature forest canopy, and to develop approaches for forecasting the positional accuracy of these harvesters under such dense canopy cover. The study aimed to fill gaps in our knowledge, such as how accurately GNSS-based harvesters can be positioned and how accurate the latest GNSS technologies are when tested in real-world forest environments, and in challenging conditions, i.e., winter operations, spruce-dominant mature stands, and high canopy cover.

This study has the potential to drive advancements in GNSS technology by providing new insights into the impact of canopy cover on GNSS's accuracy, leading to the development of GNSS technologies that are not only better suited for dense forest canopies, but are also more accurate and more efficient.

## 2. Materials and Methods

### 2.1. Study Area and Harvester Positions Data

The study was carried out in a southern boreal forest zone in Kolmikanta, Tuusniemi, Finland (62.67° N, 28.47° E, 158–162 m above sea level) (Figure 1). The study was carried out within a managed spruce-dominated stand. The average diameter of the trees was 25.2 (±5.86) cm, median 26 cm. The range was 16.5–44.5 cm. The average height was 21.6 (±2.7) m, median 22 m. The range was 16–26 m. The stand underwent mechanized commercial thinning by an eight-wheeled Ponsse Ergo cut to length (CTL) harvester.

The Trimble R12 GNSS receiver was used to collect multi-constellation GNSS data. Leica Nova TS60 robotic total station was used to collect the true positions of the harvester. The Trimble R12 and a prism for total station tracking were installed on the roof of the Ponsse Ergo harvester (Figure 2). The true position of Trimble R12 was calculated by measuring the offset between the prism and Trimble R12. The exact time based on GNSS signals was used to synchronize the measurement time between the total station and the Trimble R12. The Trimble R12 measurements were carried out in RTK mode using the

Trimnet network of reference stations available in Finland. The technical characteristics of the Trimble R12 are presented in Table 1. The accuracy numbers in the specifications only apply in open-sky conditions with nothing blocking the satellite signals.

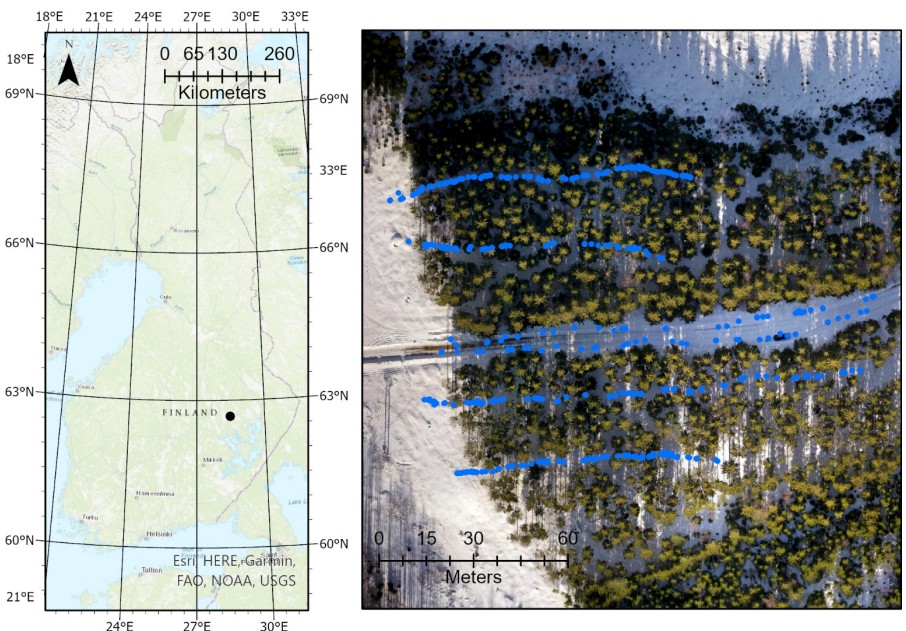

**Figure 1.** Map of the study area and established test plot with measured harvester positions. Background: unmanned aerial vehicle (UAV) and red, green, blue (RGB) image mosaic.

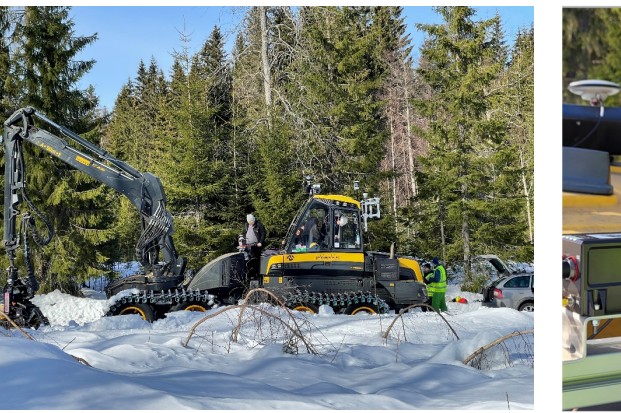
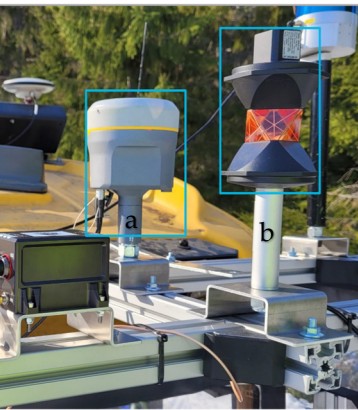

**Figure 2.** The Ponsse Ergo harvester and the equipment installation on its roof: the Trimble R12 (a); the prism (b).

The specific settings in Trimble R12 were selected to facilitate the data collection. The following details outline the selected settings:

- Inclination correction was disabled;
- Position was measured at a frequency of 1 s;
- The altitude mask was set to 10 degrees;
- The PDOP mask was set to 25.5;
- Horizontal and vertical tolerance were set to 99,999.9, which essentially disabled these parameters;
- The R12 device incorporates a feature called "xFill", which attempts to obtain RTK correction via a satellite data link if it cannot be obtained through 2G/3G.

**Table 1.** Technical specifications of the Trimble R12 GNSS receiver.

| Technical Specification | Description |
| --- | --- |
| GNSS signals received | GPS L1, L2, L2C, L5; GLONASS L1, L2, L3; Galileo E1, E5a, E5b, E6; BeiDou B1, B2, B3; QZSS L1, L2C, L5, L6; NavIC L5 |
| Channels | 672 |
| Positioning modes | RTK, DGNSS, Static, Rapid Static, PPP |
| RTK accuracy | 8 mm + 0.5 ppm horizontal; 15 mm + 0.5 ppm vertical |
| DGNSS accuracy | 30 cm (95% confidence) |
| Time to first fix (TTFF) | Cold start: <60 s; warm start: <30 s; hot start: <10 s |
| Data update rate | Up to 20 Hz |
| Operating temperature range | −40 °C to +75 °C |
| Storage temperature range | −40 °C to +80 °C |
| IP rating | IP68 (protected against dust, sand, and temporary immersion in water) |
| Battery life | Up to 10 h (RTK Rover) |
| Bluetooth | Bluetooth 4.0 and 2.1 + EDR compliant |
| Size | 17.8 cm × 11.4 cm × 4.4 cm |
| Weight | 1.07 kg (with internal battery) |

The harvester's real positions were measured by a total station with a frequency ranging from 1 to 5 measurements per second. The Trimble R12 GNSS receiver's positions were measured with a frequency of 1 Hz. To combine the positions from the total station to the Trimble R12 GNSS receiver, the positions from the total station were averaged into 1 s intervals. Total station observations deviating more than 8 mm within 1 s interval were excluded to guarantee that averaging would not introduce an additional error to the analysis. GPS time was used to synchronize the total station and the Trimble R12 observations.

For each of the Trimble R12 positions, the following additional parameters were recorded: solution type, HDOP, VDOP, PDOP, DOP, total number of satellites and satellites from each constellation observed, GPS, and GLONAS. To calculate the absolute deviation for each GNSS position relative to the total station, the easting and northing coordinates of the total station were subtracted from the corresponding GNSS position. According to the Trimble R12 specification (Table 1), the RTK accuracy is 8 mm horizontal and 15 mm vertical. Based on this, the prism points with a deviation of more than 8 mm in an east or north position were classified as inaccurate points.

*2.2. Canopy Cover Data*

To collect information on the three-dimensional structure of the canopy cover, a DJI Mavic 2 Pro drone flight was carried out at an altitude of 80 m before the harvesting operations. The drone was additionally equipped with a TOPODRONE DJI Mavic 2 Pro PPK Upgrade Kit [22]. This setup allowed a high-precision aerial survey to be performed without ground control points (GCP); however, to calibrate the focal length of the camera and to check the quality of the positional accuracy, 12 GCP were distributed in open unobstructed locations over the study area and measured with a multiband Emlid REACH RS2+ RTK GNSS receiver [23] with a 1 cm accuracy. The precise coordinates of the image centers were obtained in post-processed kinematic (PPK) mode after the images and logs from the TOPODRONE DJI Mavic 2 Pro PPK Upgrade Kit were combined in TOPODRONE post-processing software [24].

The double-grid flight pattern was employed using a consumer-grade UAV (DJI Mavic 2 Pro, Shenzhen, Guangdong, China) equipped with a Hasselblad L1D-20c RGB camera. The flights had an 80% overlap, 70% sidelap, and a camera angle of 90 degrees. The flight was automated using the Pix4Dcapture application. The unmanned aerial vehicle (UAV) data were processed in Agisoft Metashape Professional software [25] using a structure-from-motion (SfM) workflow, which is a photogrammetric technique for estimating three-dimensional structures from two-dimensional image sequences. The images were aligned, and the GCP coordinates were imported from TOPODRONE Post Processing [24]. We performed bundle adjustment in the selected coordinate system

(EPSG::3903: ETRS89/TM35FIN(N,E) + N2000 height) and built a dense point cloud with default values of "Ultra high quality" and "mild" depth filtering. The total error of the image block georeferencing was 7.76 cm. The spatial resolution of the digital surface model (DSM) was 2.7 cm. Points were classified as either "ground" or "other"; then, the digital elevation model (DEM) and the DSM (Figure 3) were constructed. Further, subtracting the DEM from the DSM allowed the canopy height model (CHM) to be calculated.

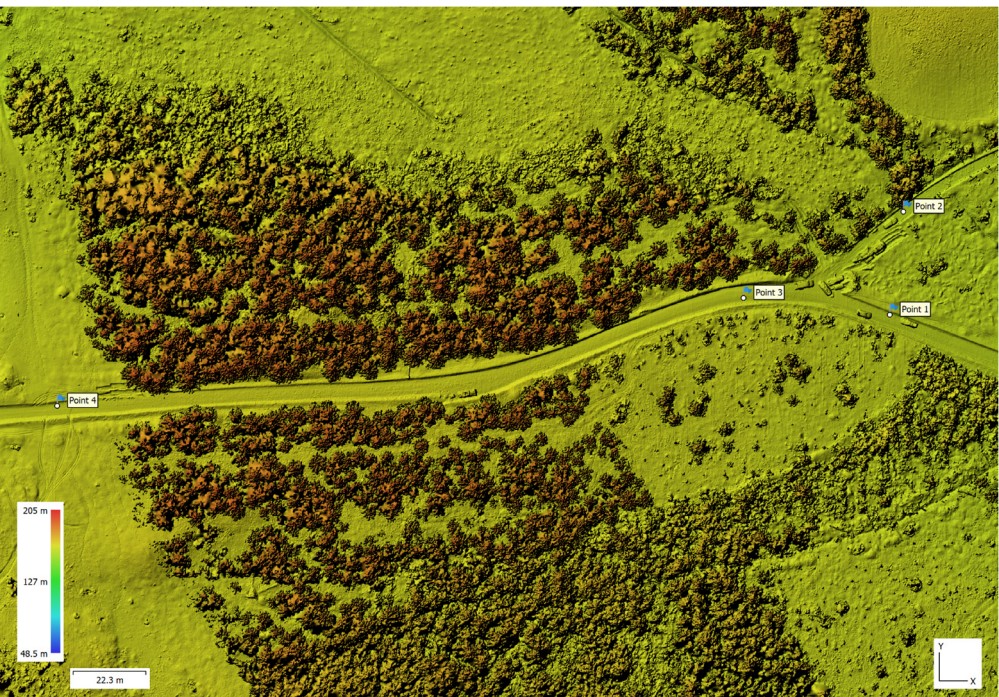

**Figure 3.** The digital surface model reconstructed using structure-from-motion process and several ground control points.

### 2.3. Analysis of Harvester Positions and Canopy Cover

To estimate the percentage of the canopy above the harvester, a buffer zone of 10 m for each of the positions was calculated in Arcgis Pro. To evaluate the extent to which the canopy cover affected the positional accuracy of the Trimble R12, the CHM was recalculated, excluding 4 m from the lower part of the canopy. The canopy cover was calculated as a percentage of the area covered by pixels above 4 m in the CHM within a 10 m radius from the harvester position point.

Viewshed analysis [26,27] was used to evaluate the impact of canopy cover on the GNSS's positioning accuracy by determining the areas where the GNSS signals were obstructed by the canopy. The analysis was carried out separately for each position in Arcgis Pro, using the harvester positions and the DSM as the input data. Analyzing the viewshed from the GNSS's receiver positions made it possible to identify the areas where the line-of-sight to the satellite was blocked by the canopy. These areas were then used to evaluate the impact of the canopy on the GNSS's accuracy by comparing the GNSS's signal quality in the obstructed areas to the signal quality in areas without canopy obstruction. This allowed the extent to which the canopy cover reduced the accuracy of the GNSS measurements to be determined. For this purpose, a 10 m radius around the observation point was used to calculate the percentage of the sky that was visible from the receiver vantage point. The results were used to compare against the positioning accuracy and to identify the areas where the GNSS signals had been blocked or degraded by the forest canopy.

## 3. Results

*3.1. Assessment of the Positioning Accuracy of the Trimble R12 GNSS Receiver Mounted on a Harvester*

The easting and northing positions obtained from the Trimble R12 GNSS receiver were compared with the true positions obtained from the total station, which allowed for the calculation of deviations. As stated, the Trimble R12 has a maximum horizontal precision of 8 mm, according to its product specification [28]. The 8 mm threshold was thus used to classify the positions as either 'Accurate' or 'Inaccurate'. From the total of 5564 observations obtained, 751 were identified as 'Inaccurate', meaning that the Trimble R12 was able to achieve the maximum horizontal precision of 8 mm for the remaining 4813 observations (86% of all observations).

The spatial distribution of the classified positions is illustrated in Figure 4, while Table 2 displays the geometric quality of the GNSS signals used to determine the positions. A further analysis of the geometric quality of the GNSS signals showed that, on average, fewer satellites were available to determine the positions for the 'Inaccurate' class positions. A paired-samples *t*-test was conducted to compare differences in the number of satellites, GDOP, PDOP, VDOP, and HDOP for 'Accurate' and 'Inaccurate' positions. There was a significant difference in the number of satellites, GDOP, PDOP, VDOP, and HDOP for 'Accurate' and 'Inaccurate' positions. These results suggest that the numbers of satellites were higher for the 'Accurate' positions than for the 'Inaccurate' positions. The values of GDOP, PDOP, VDOP, and HDOP were smaller for the 'Accurate' positions than for the 'Inaccurate' positions. This disparity is likely due to the canopy's impact on the GNSS equipment installed on the roof of the harvester. The results indicate that the Trimble R12 GNSS receiver can provide a high level of positioning accuracy with a reasonably good availability; although, the presence of a forest canopy may negatively affect its performance.

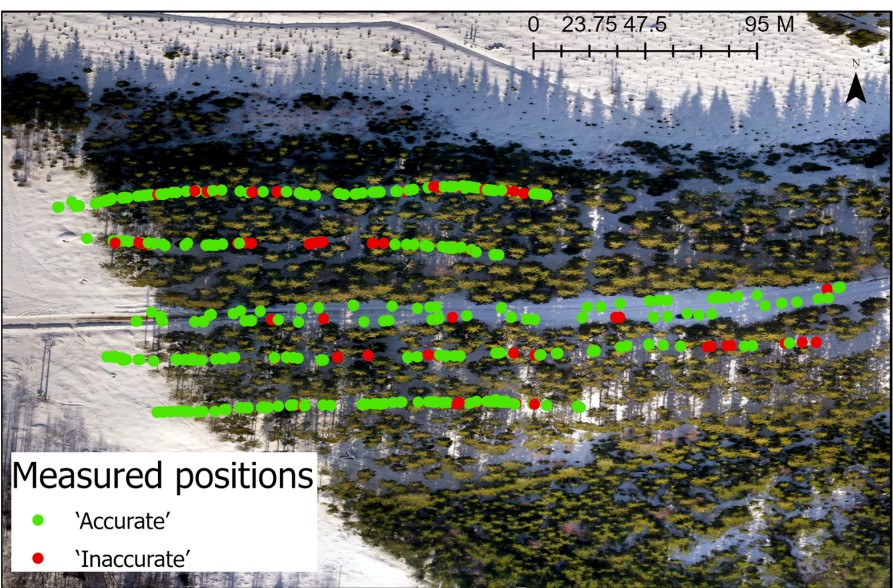

**Figure 4.** Spatial distribution of the classified measured positions using the Trimble R12 GNSS receiver mounted on a harvester.

**Table 2.** Assessment of the geometric quality of GNSS signals for determining position solutions with a Trimble R12 GNSS receiver showing significant difference from pairwise test $p < 0.001$.

| Variable | 'Accurate' (n = 4813) | | | 'Inaccurate' (n = 751) | | | t | df |
|---|---|---|---|---|---|---|---|---|
| | Mean | Std. dev. | Range | Mean | Std. dev. | Range | | |
| Number of GLONASS satellites | 3.90 | 1.27 | 0–7 | 3.23 | 1.07 | 1–6 | 15.30 | 1103 |
| Number of GPS satellites | 7.35 | 1.59 | 2–10 | 6.93 | 1.54 | 2–10 | 6.85 | 1015 |
| Number of all satellites | 11.25 | 2.26 | 5–17 | 10.17 | 1.98 | 5–14 | 13.62 | 1079 |
| Geometric dilution of precision (GDOP) | 3.00 | 1.42 | 1.6–17.3 | 3.47 | 1.98 | 1.8–16.5 | −6.81 | 875 |
| Position dilution of precision (PDOP) | 2.25 | 1.00 | 1.2–12 | 2.60 | 1.38 | 1.4–11.9 | −6.79 | 877 |
| Vertical dilution of precision (VDOP) | 1.96 | 0.88 | 1–10.9 | 2.29 | 1.24 | 1.2–11.3 | −7.00 | 870 |
| Horizontal dilution of precision (HDOP) | 1.07 | 0.52 | 0.6–9.1 | 1.19 | 0.65 | 0.7–6.4 | −4.95 | 907 |

*3.2. The Effect of Canopy Cover on the Position Accuracy of the Trimble R12 as Evaluated by Analyzing the Canopy Height Model*

The impact of canopy cover on the positional accuracy of the Trimble R12 was evaluated using a CHM. The CHM was reclassified to exclude pixels located below the Trimble R12 by using the height of the harvester roof as a reference (Figure 5). The results show that the mean canopy cover was higher for 'Inaccurate' points than for 'Accurate' points (Table 3). However, there were points that were correctly positioned even with the presence of a high level of canopy cover (86%).

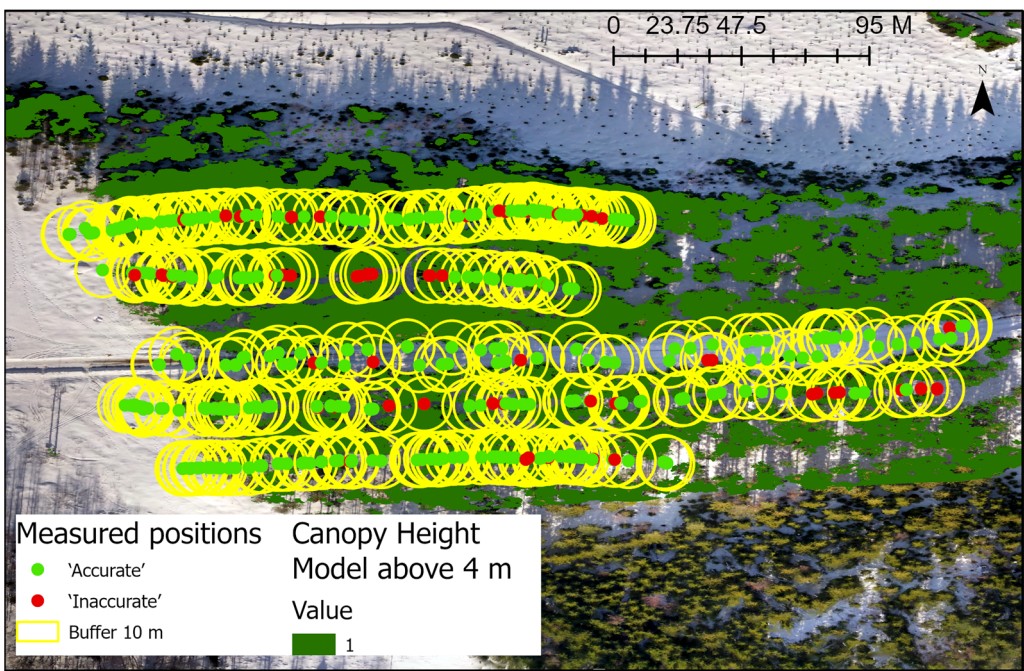

**Figure 5.** Canopy cover around the measured positions.

**Table 3.** The impact of canopy cover, estimated from the canopy height model for 'Accurate' and 'Inaccurate' harvester positions.

| Canopy Cover | 'Accurate' (n = 4813) | 'Inaccurate' (n = 751) |
|---|---|---|
| Mean | 39.51% | 62.75% |
| Minimum | 0.54% | 23.16% |
| Maximum | 86.89% | 85.78% |

A paired-samples *t*-test was conducted to compare differences in the canopy cover estimated from the canopy height model for 'Accurate' and 'Inaccurate' positions. There was a

significant difference in the canopy cover estimated from the canopy height model for 'Accurate' positions (M = 39.51%) and 'Inaccurate' positions (M = 62.75%); t(1217) = −34.46, *p* = 0.000).

A simple linear regression was performed to predict the PDOP based on the canopy cover above the roof level of the harvester, as estimated from the CHM. The regression equation was found to be significant (F(1,5563) = 15,499,106, *p* < 0.000), with an $R^2$ of 0.735. The equation: PDOP = 0 + 4.309 ∗ canopy cover, %.

Another regression was calculated to predict the maximum possible deviation from the total station in northing or easting directions, as estimated by the Trimble R12 based on the canopy cover above the roof of the harvester, calculated based on the CHM. The regression equation was found to be significant (F(1,5563) = 5,512,439, *p* < 0.000), with an $R^2$ of 0.497. The equation: maximum deviation = 0 + 0.733 ∗ canopy cover, %. This means that the maximum deviation from the true position increased by 0.733 cm for each percentage increase in the canopy cover.

*3.3. The Impact of Canopy Cover on the Positional Accuracy of the Trimble R12 as Evaluated by Conducting a Viewshed Analysis of the Digital Surface Model*

In order to take into account the specific factors influencing how visible the satellites are from a GNSS device vantage—differences in elevation, variations in tree height, species variations, crown size variations, and distances to many trees of different forms and sizes—a viewshed analysis of the DSM was carried out for each point (Figure 6, Table 4). The individual height of each point was used to calculate the viewshed within the DSM. Due to the complexity of the calculations (129 h for 1879 positions on 12th Gen Intel(R) Core(TM) i9-12900HK 2.90 GHz with NVIDIA GeForce RTX 3080 Ti Laptop GPU), a viewshed analysis was carried out for 573 randomly selected 'Inaccurate' and 1306 randomly selected 'Accurate' points.

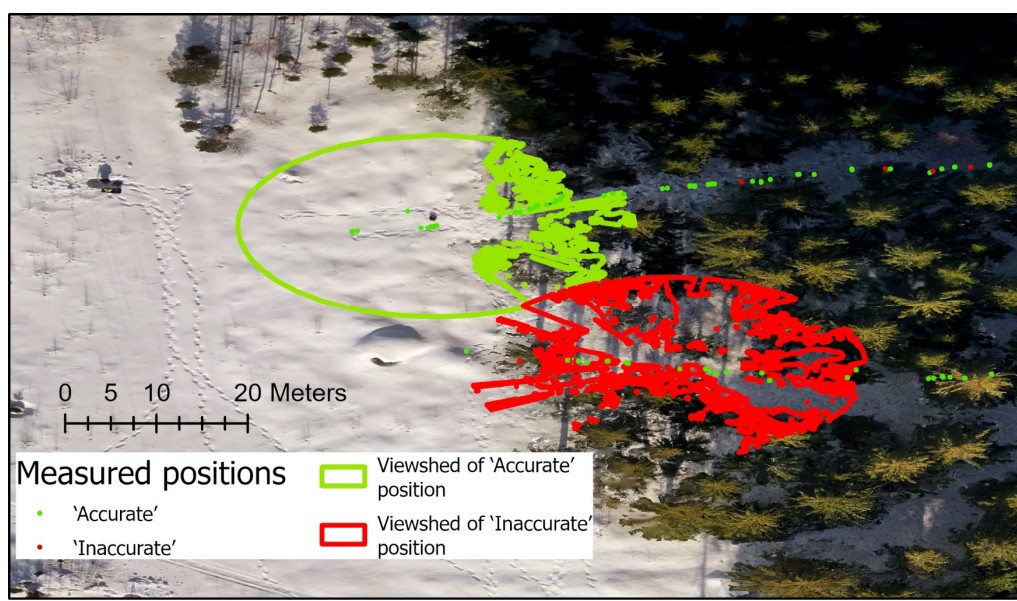

**Figure 6.** Examples of the viewshed analyses for 'Accurate' and 'Inaccurate' points.

**Table 4.** The impact of canopy cover estimated from viewshed analysis of DSM for 'Accurate' and 'Inaccurate' positions.

| Canopy Cover | 'Accurate' (n = 1306) | 'Inaccurate' (n = 573) |
|---|---|---|
| Mean | 51.50% | 61.69% |
| Minimum | 4.03% | 24.40% |
| Maximum | 100.00% | 95.20% |

A paired-samples *t*-test was conducted to compare differences in the canopy cover estimated by the viewshed analysis of the DSM for 'Accurate' and 'Inaccurate' positions. There was a significant difference in the canopy cover estimated from the canopy height model for 'Accurate' positions (M = 51.50%) and 'Inaccurate' positions (M = 61.69%); t(903) = −37.18, *p* = 0.000).

A simple linear regression was performed to predict the PDOP based on the canopy cover estimated by the viewshed analysis of the DSM. The regression equation was found to be significant (F(1,1879) = 5,508,671, *p* < 0.000), with an $R^2$ of 0.745. The equation was PDOP = 0 + 4.132 ∗ canopy cover, % from the viewshed analysis of the DSM.

Another regression was calculated to predict the maximum possible deviation from the total station in northing or easting directions, as estimated by the Trimble R12 based on the canopy cover estimated by the viewshed analysis of the DSM. The regression equation was found to be significant (F(1,1879) = 2,424,240, *p* < 0.000), with an $R^2$ of 0.563. The equation was maximum deviation = 0 + 0.774 ∗ canopy cover %, from the viewshed analysis of the DSM. This means that the maximum deviation from the true position increased by 0.774 cm for each percentage increase in the canopy cover estimated from the viewshed analysis.

To predict the number of satellites that were visible to the Trimble R12 based on the canopy cover estimated in the viewshed analysis of the DSM, we performed another regression. The regression equation was found to be significant (F(1,1879) = 7,070,107, *p* < 0.000), with an $R^2$ of 0.790.

## 4. Discussion

The results suggest that the Trimble R12 GNSS receiver, when mounted on a harvester, is capable of achieving a high level of positioning accuracy and availability, with 86% of the observations meeting the maximum horizontal accuracy of 8 mm. The spatial distribution of the classified positions is shown in Figure 4, while Table 2 provides an overview of the geometric quality of the GNSS signals used in determining the positions. The analysis of the geometric quality of the GNSS signals showed that the presence of a canopy impacts the performance of the Trimble R12, as indicated by larger values of GDOP, PDOP, VDOP, and HDOP. Despite these challenges, the RTK corrected Trimble R12 GNSS receiver mounted on the roof of the harvester still demonstrates a high level of positioning accuracy.

The results of this study indicate that the existence of canopy cover has a significant effect on the positional accuracy of the Trimble R12 device. The mean canopy cover for 'Inaccurate' points was found to be higher (62.75%) than that for 'Accurate' points (39.51%). However, it is also noted that there were points with a correct positioning even with a high level of canopy cover of 86%.

The study used a CHM to evaluate the impact of canopy cover on the positional accuracy. In this study, dense point clouds for the CHMs creation were acquired with UAV imagery, which may be laborious in operational use. Many countries have national laser scanning data available at point densities from 0.5 to several points per square meter, and further studies are needed to estimate how well these data can serve as a source for canopy cover and viewshed analysis.

A simple linear regression was performed to predict the PDOP and the maximum possible deviation from the total station in northing or easting directions, both based on the canopy cover estimated from the CHM. Both regression equations were found to be significant with high $R^2$ values (0.735 for PDOP and 0.497 for maximum deviation). This indicates that the maximum deviation from the true position increased by 0.733 cm for each percentage increase in the canopy cover. These results suggest that canopy cover has a significant impact on the positional accuracy of the Trimble R12 device. It is thus necessary to consider this factor when using the device, and similar alike, in forestry operations under canopy cover.

The study analyzed the impact of canopy cover on the positional accuracy of the Trimble R12 using a viewshed analysis of a DSM. The results showed that 'Inaccurate' points had a higher mean level of canopy cover than the 'Accurate' points. The canopy

cover estimated from the viewshed analysis of the DSM was found to be a significant predictor of the PDOP and the maximum possible deviation from the true position in northing or easting directions. In addition, the number of satellites that were visible to the Trimble R12 was found to be significantly predicted by the percentage of canopy cover estimated through the viewshed analysis of the DSM. However, we did not analyze the visibility of each satellite available in the sky, and the consequential satellite geometry, at the time of the position observation. Such an analysis may provide further insight into the positioning degradation.

The regression analysis revealed that the viewshed analysis of the DSM has a stronger relationship with accuracy ($R^2$ = 0.563) compared to the relationship between accuracy and the analysis of the canopy cover model (0.497). However, it should be noted that in the analysis of the canopy cover model, the potential impact of terrain slope and shadow on satellite visibility was not taken into consideration. While this factor may not have a significant effect on the specific environment under investigation, it could be of great importance in the broader context of the global industry. To address this limitation, we propose the use of the whole digital surface model (DSM) for evaluating the impact of canopy cover on accuracy. This approach considers the terrain slope and shadow effects on satellite visibility, providing a more comprehensive and accurate analysis of the canopy cover model. Similarly, a stronger relationship was observed for the viewshed analysis of the DSM in relation to PDOP. Despite the greater computational demands of the viewshed analysis of the DSM, these findings suggest that it provides more precise results when evaluating the impact of canopy cover on the harvester GNSS's positioning.

The results of this study are comparable with a similar study [10], where the determination coefficient ($R^2$) in the regression analysis for a coniferous forest model was 0.579. This indicated that the model could predict the maximum positioning error under different canopy covers.

In our study, 86% of observations had a maximum horizontal precision of 8 mm, a percentage which obviously may vary in different forest environments. Integrating the GNSS with LiDAR could improve the accuracy by compensating for the errors introduced by the canopy cover. Advanced GNSS technologies can benefit also from integration with GNSS-aided inertial navigation systems. Continuous cover forestry is expected to decrease the impact of canopy cover on the GNSS's accuracy.

This study has various limitations. For instance, some trees might have been harvested before collecting the canopy cover data, which could have underestimated the impact of canopy cover on the observed GNSS signals. The terms accuracy and precision were used, but it should be noted that the precise positions were not necessarily accurate. The study focused on PDOP and assumed that this was directly related to accuracy, but we also acknowledge that this was not necessarily true. It was possible to obtain precise clusters of positions that were inaccurately located. In forest environments, there were challenges with GNSS's positioning due to signals from multiple satellites reaching the receiver via indirect paths, resulting in long periods of signal interruptions and reflections. Furthermore, the boom on the harvester could have potentially caused problems for the GNSS signals as it passed close to the receiver, or when a tree held in the processor head passed close to the receiver. This could have caused signal interruptions and/or reflections for short durations, which were often not filtered out by the receiver, leading to position errors.

The models to predict PDOP developed within this study have limitations as they were forced to zero intercept due to the reason that there were no points available within open-sky conditions. The minimum canopy cover observation from the canopy height model was 0.54%, while the minimum canopy cover evaluated by the viewshed analysis of the DSM was 4.03%. When looking at the observed minimum for PDOP, it was found to be approximately 1.2 for accurate points and 1.4 for inaccurate points. It is possible that even lower values for PDOP could be expected in open-sky conditions, but, unfortunately, within this study, there were no points available without the impact of canopy cover. Future studies should investigate this further.

This study also has a limitation in terms of the radius used to assess the impact of the canopy on GNSS signals. The 10 m radius was probably considered too small to capture the full extent of the impact. The trees in the area had an average height of 22 m with a radius of 10 m, and the GNSS's antenna was 4 m above the ground; satellites below about 62 degrees were not taken into consideration. This could have potentially led to an underestimation of the amount of canopy that impacts signals from several satellites that were below this angle.

Future studies should repeat the test without harvesting and compare the results before and after harvesting. Future research on the impact of canopy cover on harvester positioning under the canopy should consider the following investigations:

1.  Conduct long-term field studies to monitor the GNSS's accuracy over time under different weather and foliage conditions;
2.  Investigate the use of GNSS signal enhancement techniques—for example, GNSS reflectometry—to improve the accuracy under canopy cover;
3.  Apply machine learning algorithms to GNSS data to predict the accuracy based on foliage density and tree height, and evaluate the effectiveness in real-world applications;
4.  Investigate integrating the GNSS with other sensors, such as LiDAR and cameras, to improve accuracy under canopy cover;
5.  Use a priori forest models/tree maps, e.g., from ALS and integrated LiDAR SLAM, to replace the need for a total station for practical long-term studies and forestry practice (1).

## 5. Conclusions

The results suggest that the Trimble R12 GNSS receiver, when mounted on a harvester, is capable of achieving a high level of positioning accuracy, with 86% of the observations meeting the maximum horizontal precision of 8 mm. The analysis of the geometric quality of the GNSS signals showed that the presence of a canopy may impact the performance of the Trimble R12; this is evidenced by the lower number of satellites used in determining 'Inaccurate' positions and the higher values of GDOP, PDOP, VDOP, and HDOP. Canopy cover estimated from the viewshed analysis of the DSM was found to be a significant predictor of PDOP and the maximum possible deviation from the true position in northing or easting directions. Despite the greater computational demands of the viewshed analysis of the DSM, these findings suggest that it provides more precise results when evaluating the impact of canopy cover on the harvester GNSS's positioning.

**Author Contributions:** Conceptualization, K.V. and L.S.; methodology, E.L.; formal analysis, E.L., A.K., H.K. and E.H.; investigation, E.L.; resources, L.S., J.R. and Y.N.; writing—original draft preparation, E.L.; writing—review and editing, E.L., K.V., A.K., H.K., E.H., L.S., J.R. and Y.N.; visualization, E.L.; project administration, K.V.; funding acquisition, J.R. and J.H. All authors have read and agreed to the published version of the manuscript.

**Funding:** This research was funded by the Academy of Finland Flagship Programme (Forest-Human–machine Interplay (UNITE)) [grant No 337653], Density4Trees [Decision No 331708], the Natural Resources Institute Project "CCFBASIS", and project BOFORI (KA4002) within the Karelia CBC Programme financed by the European Union.

**Institutional Review Board Statement:** Not applicable.

**Informed Consent Statement:** Not applicable.

**Data Availability Statement:** The data presented in this study are available on request from the corresponding author.

**Acknowledgments:** The authors wish to thank Metsä Group, Ponsse Plc, and Motoajo Ltd. for help and collaboration in collecting the measurements for this work.

**Conflicts of Interest:** The authors declare no conflict of interest. The funders had no role in the design of the study; in the collection, analyses, or interpretation of the data; in the writing of the manuscript; or in the decision to publish the results.

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
