# Peer review of "Unlocking Digitalization in Forest Operations with Viewshed Analysis to Improve GNSS Positioning Accuracy"

_forests, doi:10.3390/f14040689_

Round 1

Reviewer 1 Report

General comments

This manuscript needs a thorough review for use of English and sentence structure. There are several sentences where the structure is awkward making it hard to understand the meaning.

You use the terms accuracy and precision interchangeably in many parts of the manuscript. Precise positions are not necessarily accurate. You focus you analyses on PDOP assuming that this is directly related to accuracy but this is not necessarily true.

Do you think the boom on the harvester causes any problem for the GNSS signals? It looks like the boom could pass close to the GNSS receiver or a tree held in the processor head could pass close to the GNSS receiver. This could cause signal interruptions and/or reflections for short durations which are often not filtered out by the receiver leading to position errors.

Lines 211-212: You need to determine (paired t-test) whether differences in GDOP, PDOP, VDOP, and HDOP are statistically significant between accurate and inaccurate positions. Looking at the means and ranges in table 2, I’m not sure that these values are significantly different. If the differences are significant, the rest of your analyses hold up. However, if differences are not significant, all your efforts to predict PDOP from direct canopy cover or cover by viewshed as a surrogate for accuracy are not meaningful. Even if you have a strong relationship between canopy cover and PDOP, this doesn’t tell you if a position is “accurate”. Similar issue with the cover values presented in table 4…are these significantly different? In line 298, you conclude that canopy cover has a significant influence on accuracy but I’m not sure your statistical analyses justify this statement. You present “significant regressions” as evidence that canopy cover is related to PDOP. However, you have not established that PDOP is a good predictor of accuracy.

I would argue that the relationship between PDOP and position accuracy is not well known in difficult conditions. PDOP is related to precision (or repeatability) and not directly to accuracy. It is possible to have tight (precise) clusters of positions (1 second epochs) that are in the wrong place (inaccurate). This occurs in forest conditions and urban conditions (urban “canyons”) when you have long periods where signals from several satellites reach the receiver via an indirect path.

When fitting models to predict PDOP, it appears that you forced a zero intercept. Is this realistic? If canopy cover is 0, do you expect PDOP = 0? The observed minimum for PDOP was ~1.2 for accurate points and 1.4 for inaccurate points. Would you expect even lower values for PDOP in open-sky conditions?

Comments referenced to line numbers

Lines 48-49: it isn’t clear how inaccurate positions lead to changes in the carbon footprint

Lines 62-63: it isn’t clear in this sentence if continuous cover forestry or conventional even aged forest management ends with clearcutting. Maybe change to something like this:

Conventional even aged rotational forest management ends in clear cutting and forest regeneration by natural, sowing or planting [13–15]. Continuous cover forestry is a silvicultural alternative aimed at achieving multifunctional objectives including increased species diversity, more resilient stand structure, and increased carbon storage [16–21].

Lines 87-90: While this could be a true statement, are you working with manufacturers of GNSS receivers to improve their products? My experience with survey-grade GNSS receivers leads me to believe that most (maybe all) manufacturers are focused more on accuracy under good conditions and less on accuracy under difficult conditions. While there has been work on processing algorithms to deal with non-line-of-sight and multi-pathing problems, these problems will always exist in forest conditions. I see more promise for other positioning technologies such as ultra-wideband positioning systems or systems that combine GNSS and IMU technologies.

Lines 97-98: height units should be meters. The second sentence in line 98 is redundant.

Table 1: it is probably worth mentioning in the text that the accuracy numbers in the specifications only apply in open-sky conditions with nothing blocking the satellite signals.

Lines 141-143: this sentence is unclear. What is being subtracted from what? It sounds like you are computing the GNSS position relative to the total station but I’m not sure. Also not sure what absolute deviation refers to.

Line 144: Table 1 mentions 8 and 15 mm accuracy but you refer to precision in this sentence. These are not the same thing.

Lines 153-155: Were the GCPs in open, unobstructed locations or under canopy? If under canopy, the accuracy for the Emlid receiver will not be 1cm. Figure 3 show GCPs but not all GCPs. I suggest and additional sentence describing the conditions are the GCPs.

Line 171: “extracting” should be “subtracting”

Lines 177-178: this sentence is a bit confusing because you mention the level of the harvester roof and then describe a 10m buffer (assume a horizontal distance). I think you could just say “percentage of the canopy above the harvester”

Line 192: It seems that the 10m radius is too small to assess the full impact of the canopy on GNSS signals. You have trees averaging 22m in height and with a 10m radius and assuming the GNSS antenna is 3m above the ground you would not consider the amount of canopy for satellites below about 62 degrees (atan(19/10)). I would expect several satellites below this angle so you are underestimating the amount of canopy that could impact signals from these satellites.

Lines 212-213: See the general comment regarding the effect of the boom and/or a tree held in the boom on GNSS signals.

Lines 241-242 & line 247, 270, 276, 283: This sentence isn’t clear. What is “0 + 4.309”? You explain this in lines 249-250 but you need to fix the sentences to show the full equation. For example PDOP = 0 + 4.309 * CanopyCover.

Lines 268-269: what does “closing the sky direct for satellite signals” mean?

Lines 283-284: I’m not sure this equation makes sense. If the equation is #sats = 0 + 16.58*cover, under open sky (cover = 0), there would be no satellites visible. Maybe this is actually the equation for predicted maximum deviation…not clear.

Line 289: do you mean accuracy instead of precision?

Line 312-314: You need to resent the equation if you are referencing the coefficient for cover (0.733).

Reviewer 2 Report

The paper entitled "Unlocking Digitalization in Forest Operations with Viewshed 2 Analysis to Improve GNSS Positioning Accuracy" is a piece of research that presents an assessment of the potential accuracy of operating harvesters based on modern GNSS devices. The manuscript exhibits many data and models relevant to science, especially Forestry. The methodology section is significant and detailed, and the results are presented in a high-quality way. The aim and methods are in line with the scope of the journal. Moreover, the paper contributes to a steamily well-developed investigation in the field of Forest studies. Therefore, I recommend some minor revisions before its publication in the journal.

Figure 3 lacks a coordinate system and a north arrow. The letters and annotations need to be larger.

Figure 4 needs to change the scale bar to "m" instead of the word "Meters." Both Figure 4 and Figure 5 need to include a north arrow. The same issue occurs in Figure 6.

Did the authors conduct experiments with L-band satellite images? The results could be improved by combining a LIDAR camera with L-Band Satellite images such as SAOCOM from CONAE, Argentina.

Author Response

Dear Anonymous Reviewer,

We would like to express our sincere gratitude for the time and effort you have taken to review our manuscript titled "Unlocking Digitalization in Forest Operations with Viewshed Analysis to Improve GNSS Positioning Accuracy" submitted to Forests journal. Your feedback and comments have been invaluable in improving the quality and clarity of our paper.

We appreciate your comments regarding the value and relevance of our research. We also thank you for the constructive criticism and suggestions that you provided. Your feedback has been extremely helpful in identifying areas that required further development and refinement.

We are pleased to inform you that we have carefully considered and implemented the majority of your comments, which have significantly improved the language and style of our manuscript. The paper underwent significant editing to improve its English language and style. We have also taken into account your suggestions regarding specific sections of the paper, and we have made the necessary revisions to improve the clarity and accuracy of our results and conclusions. Please find below our responses to your specific comments:

Point 1: Figure 3 lacks a coordinate system and a north arrow. The letters and annotations need to be larger.

Response 1: Regarding your comment on Figure 3, we want to inform you that the figure was produced using Agisoft Metashape software, which automatically generates the output without a coordinate system or a north arrow. Unfortunately, we cannot modify these aspects of the figure, as they are inherent to the software output.

Point 2: Figure 4 needs to change the scale bar to "m" instead of the word "Meters." Both Figure 4 and Figure 5 need to include a north arrow. The same issue occurs in Figure 6.

Response 2: Corrected as suggested by reviewer.

Point 3: Did the authors conduct experiments with L-band satellite images? The results could be improved by combining a LIDAR camera with L-Band Satellite images such as SAOCOM from CONAE, Argentina.

Response 3: Thank you for your question. Based on our study, we did not conduct experiments with L-band satellite images, but we agree that the combination of LIDAR camera with L-band satellite images, such as SAOCOM from CONAE, Argentina, could be a promising direction for future research to improve the results. We appreciate your suggestion and will keep it in mind for future studies.

Once again, we would like to express our gratitude for your thoughtful review and constructive feedback. We believe that your comments have greatly enhanced the overall quality of our paper, and we are confident that the revised version will meet the high standards of the Forests journal.

Sincerely,

Eugene Lopatin

Reviewer 3 Report

Accurate positioning is important for forest digitalization and operations. Here Lopatin et al., evaluated the GNSS positioning accuracy on forest harvesters and explored the impacts of canopy cover estimated from viewshed analysis on the accuracy. The innovation of this paper is okay for me but substantial revisions are needed before I would recommend it for publication.

Specific comments:

1.      Title: This study just evaluated the GNSS accuracy and explored its potential drivers that impact the positioning accuracy at a forest instead of improving the GNSS accuracy and unlocking digitalization. Therefore, please revise the title to make it exactly reflect what you did in the main text.

2.      Line 178, why 10 m buffer was selected, any reason? Does this buffer setting affect the results a lot?

3.      Section 3.2, what about the impacts of canopy cover on GDOP, VDOP, and HDOP in addition to PDOP.

4.      Section 3.3, what about the impacts of canopy cover estimated by viewshed analysis of the DSM on GDOP, VDOP, and HDOP in addition to PDOP.

5.      Line 292-295: reduced number of satellites available and larger values of errors for “inaccurate” positions are not necessarily due to the presence of the canopy, or results in section 3.1 cannot reflect the impacts of canopy.

6.      Did the number of available satellites affect the positioning accuracy more or the canopy cover affect the accuracy more? Perhaps, a random forest model could link those different predictors with the positioning bias and give the importance of each predictor.

7.      Substantial improvement in the writing is needed. Here I listed several examples as bellows but please revise all through the paper.

8.      Line 24: “was found a significant predictor” to “was found as a significant predictor”

9.      Line 34: delete “is a vital industry that”

10.   Line 47: “yield reduced negative … impacts”? I am not sure what do you specifically mean here.

11.   Line 62-65: too long a sentence and seems less understandable. Please rewrite it.

12.   Line 375: “is evidence” to “is evidenced by”

Round 2

Reviewer 1 Report

The manuscript is much improved. The addition of the t-tests strengthens the analyses and presentation.

You need to review the manuscript for consistent use of the decimal separator. You have some (most) numbers using “,” but a few that use “.”.

Author Response

Thank you for review.  The consistent use of the decimal separator was checked and corrected.

Reviewer 3 Report

Thanks for the authors' efforts on addressing my concerns. Current version is okay for publication.

Author Response

Thank you for review!